

# Response of Bolivian gray titi monkeys (*Plecturocebus donacophilus*) to an anthropogenic noise gradient: behavioral and hormonal correlates

Lucero M. Hernani Lineros[1,2,*], Amélie Chimènes[3], Audrey Maille[3], Kimberly Dingess[4], Damián I. Rumiz[1,5] and Patrice Adret[1,*]

[1] Zoología Vertebrados, Museo de Historia Natural Noel Kempff Mercado, Santa Cruz de la Sierra, Bolivia
[2] Carrera de Biología, Universidad Mayor de San Andrés, La Paz, Bolivia
[3] Unité Eco-anthropologie UMR 7206, Museum National d'Histoire Naturelle, CNRS, Université de Paris, Paris, France
[4] Marshall University, Huntington, WV, United States of America
[5] Fundación Simón I. Patiño, Santa Cruz de la Sierra, Bolivia

[*] These authors contributed equally to this work.

Corresponding author
Patrice Adret,
padret@museonoelkempff.org

## ABSTRACT

Worldwide urban expansion and deforestation have caused a rapid decline of non-human primates in recent decades. Yet, little is known to what extent these animals can tolerate anthropogenic noise arising from roadway traffic and human presence in their habitat. We studied six family groups of titis residing at increasing distances from a busy highway, in a park promoting ecotourism near Santa Cruz de la Sierra, Bolivia. We mapped group movements, sampled the titis' behavior, collected fecal samples from each study group and conducted experiments in which we used a mannequin simulating a human intrusion in their home range. We hypothesized that groups of titi monkeys exposed to higher levels of anthropogenic noise and human presence would react weakly to the mannequin and show higher concentrations of fecal cortisol compared with groups in least perturbed areas. Sound pressure measurements and systematic monitoring of soundscape inside the titis' home ranges confirmed the presence of a noise gradient, best characterized by the root-mean-square (RMS) and median amplitude (M) acoustic indices; importantly, both anthropogenic noise and human presence co-varied. Study groups resided in small, overlapping home ranges and they spent most of their time resting and preferentially used the lower forest stratum for traveling and the higher levels for foraging. Focal sampling analysis revealed that the time spent moving by adult pairs was inversely correlated with noise, the behavioral change occurring within a gradient of minimum sound pressures ranging from 44 dB(A) to 52 dB(A). Validated enzyme-immunoassays of fecal samples however detected surprisingly low cortisol concentrations, unrelated to the changes observed in the RMS and M indices. Finally, titis' response to the mannequin varied according to our expectation, with alarm calling being greater in distant groups relative to highway. Our study thus indicates reduced alarm calling through habituation to human presence and suggests a titis' resilience to anthropogenic noise with little evidence of physiological stress.

## INTRODUCTION

Owing to the constant expansion of cities and deforestation around the world, humans have increasingly invaded wildlife habitats, reshaping the landscape into man-altered ecosystems (*Hendry, Gotanda & Svensson, 2017*). Although urban habitats may provide benefits to city-dwelling animals, such as greater food availability and less exposure to natural predators (*Muhly et al., 2011*; *Adams, 2016*), anthropogenic disturbances often have detrimental consequences on wildlife (*Ciuti et al., 2012*; *Hendry, Gotanda & Svensson, 2017*; *Palacios, D'Amico & Bertellotti, 2018*), which has given rise to a field of research known as "urban wildlife ecology" that aims to understand the effects of human disturbance on animal populations that live within the cities or close to human settlements (*Gill, Sutherland & Watkinson, 1996*; *Frid & Dill, 2002*; *Magle et al., 2012*; *Gaynor et al., 2018*).

For many organisms, an omnipresent source of stress is represented by anthropogenic sound (*Slabbekoorn et al., 2018*; *Kunc & Schmidt, 2019*; *Raboin & Elias, 2019*), also called anthropophony (*Pijanowski et al., 2011*). Anthropophony is closely associated with human activity routine, from construction and mass recreation to transportation (aerial, terrestrial and marine), the latter being considered as one of the most pervasive acoustic perturbations on Earth (*Barber, Crooks & Fristrup, 2010*; *Shannon et al., 2016*). Anthropophony can alter the behavior, phenotype and homeostasis of animals living near or within the cities, with potential effects at a population level (*McGregor et al., 2013*; *Giraudeau et al., 2014*). Indeed, noise pollution has a dramatic impact on habitat selection, foraging patterns and communication networks of animals (anurans: *Bee & Swanson, 2007*; *Grace & Noss, 2018*; birds: *Swaddle & Page, 2007*; terrestrial mammals: *Shannon et al., 2014*; marine mammals: *Nowacek et al., 2007*; *McMullen, Schmidt & Kunc, 2014*) with serious implications for their survival and reproduction (reviewed in *Brumm & Slabbekoorn, 2005*; *Blickley & Patricelli, 2010*; *Brumm, 2013*; *McGregor et al., 2013*). Collateral nociceptive effects induced by noise include hearing impairments, cardiovascular defects, surge in glucocorticoid secretion, sleep and immune system disorders, and may also compromise DNA integrity and gene expression (*Kight & Swaddle, 2011*; *Slabbekoorn et al., 2018*).

To mitigate or counteract the stress incurred by anthropogenic noise, animals can generate different responses: fleeing as a result of fear (*Laundré, Hernández & Ripple, 2010*), showing awareness of the source of noise via increased vigilance (*Frid & Dill, 2002*; *Barber, Crooks & Fristrup, 2010*) or coping with the disturbance through habituation (*Walker, Dee Boersma & Wingfield, 2005*). In the process, many species have been shown to exhibit vocal flexibility, such as amplitude and frequency shift, in order to avoid signal masking (reviewed in *Brumm & Slabbekoorn, 2005*; *Brumm, 2013*; *McGregor et al., 2013*; *Slabbekoorn et al., 2018*; but see *Zollinger et al., 2018*). Noise tolerance is species-specific and depends more on the animals' sensory capacities and decision-making than on the intensity of sound *per se* (reviewed in *Saunders & Dooling, 2018*).

Growing concerns of the harmful impact of chronic anthropogenic noise on wildlife behavior and physiology have prompted researchers to investigate the effect of noise pollution in Neotropical primates, such as Atelidae (spider monkeys, *Ateles spp.*: *Rangel-Negrín et al., 2009*; *Rimbach et al., 2013*; *Vanlangendonck et al., 2015*; howler monkeys, *Alouatta spp.*: *Martínez-Mota et al., 2007*; *Behie, Pavelka & Chapman, 2010*; *Rimbach et al., 2013*; *Vanlangendonck et al., 2015*; *Cantarelli et al., 2017*), Pitheciidae (titi monkeys, *Callicebus nigrifrons*: *Duarte et al., 2017*) and Callithrichidae (marmosets, *Callithrix penicillata*: *Duarte et al., 2011*; *Santos et al., 2017*; tamarins, *Saguinus leucopus*: *Soto-Calderón, Álvarez Cardona & García-Montoya, 2016*). For example, black-tufted marmosets (*C. penicillata*) have been shown to avoid noisy areas of an urban park, irrespective of food availability (*Duarte et al., 2011*).

In a comprehensive review of the effects of anthropogenic noise on wildlife, *Shannon et al. (2016)* provide several important recommendations, one of which is to measure the animals' responses over a gradient of noise levels. Indeed, heavy traffic noise on major highways of city outskirts gives rise to a noticeable acoustic gradient in the landscape. Primates—and other animals—roaming in the remnant forests along these roads are thus exposed to varying levels of anthropogenic noise, depending on the distance of their home range from the road. This, in turn, may affect the monkeys' behavior and homeostasis (*Sapolsky, Romero & Munck, 2000*).

To evaluate the impact of an anthropogenic noise gradient on the behavior and physiology of a peri-urban population of titi monkeys (Pitheciidae; sub-family Callicebinae), we conducted a study integrating acoustics, ethology and non-invasive stress endocrinology in one suite. We hypothesized that family groups residing in areas with greater exposure to anthropophony will differ both behaviorally (higher tolerance of humans) and physiologically (elevated glucocorticoids) from groups least exposed to anthropophony.

## MATERIALS & METHODS

### Study area

The study was conducted in the Ecological Park of Yvaga Guazú (S14°14′30″, W66°58′39″), 13 km from downtown Santa Cruz de la Sierra (Fig. 1A). This privately-owned property of 15 ha is home to a small, but thriving population of Bolivian gray titi monkeys (*P. donacophilus*; *Wallace et al., 2018*) that share the area with several groups of Azara's owl monkeys (*Aotus azarae*) and squirrel monkeys (*Saimiri boliviensis*). The park is surrounded to the north by a busy highway and to the south by the quiet suburb of Campo Verde. The northern section of the park, where human presence abounds, consists of a cultivated garden that exhibits both native and exotic plants to promote ecotourism. Further south, the garden gives way to a remnant of Chiquitano dry forest (∼7 ha), including species such as *Ceiba speciosa*, *Cedrela odorata*, *Enterolobium contortisiliquum*, *Zeyheria tuberculosa*, *Samanea tubulosa*, *Albizia niopoides*, *Swietenia macrophylla* and *Ficus* sp. with an abundant understory of Piperacae shrubs. Some twenty employees work in the park on a daily basis (8:00–18:00 h), assuming various functions such as gardening, horticulture and general

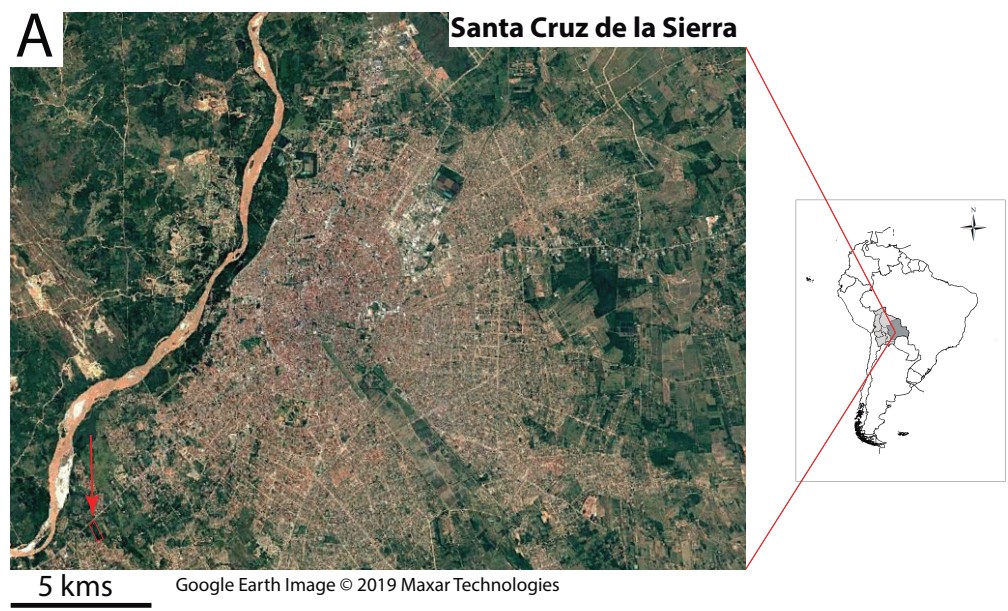

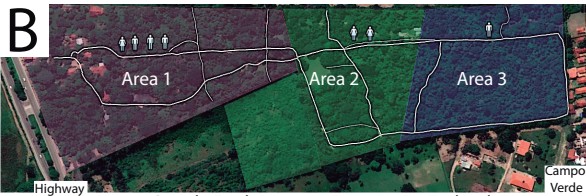

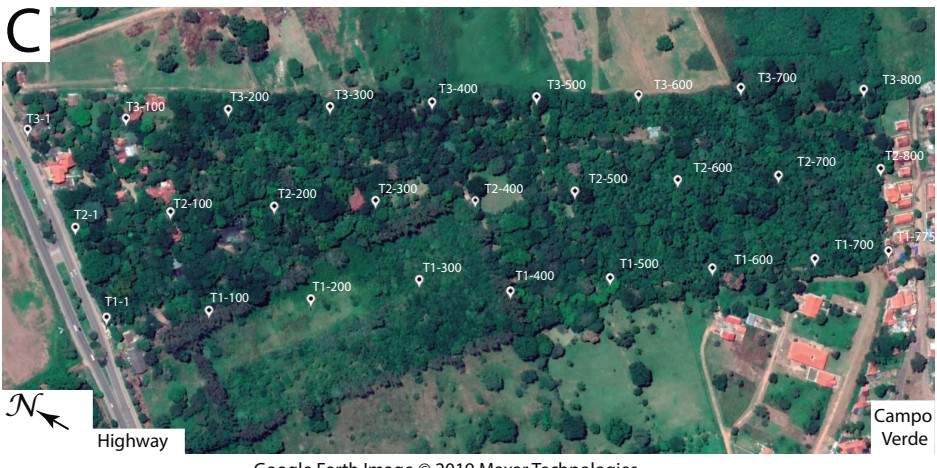

**Figure 1  Study area.** (A) Geographic location of the Ecological Park of Yvaga Guazú (red arrow pointing to red rectangle, bottom left) relative to the city of Santa Cruz de la Sierra. (B) Map sketching the three areas in the park where human disturbance was subjectively evaluated. Symbols denote a gradient in human presence from the entrance of the park to the quiet suburb of Campo Verde. (C) The grid layout consists of 27 points evenly spaced out along each of three transects (T1, T2 and T3), labeled according to their respective distance to the highway. Google Earth image ©2019 Maxar Technologies.

**Table 1  Main sources of anthropogenic disturbance heard in the study area.**

| Source | Description |
|---|---|
| Aerial traffic | Helicopter, small plane, jet aircraft |
| Roadway traffic | Vehicles, horn, siren |
| Machinery | Tractor, lawnmower |
| Tools | Machete, sprinkler, ladder, hammer |
| Recreation | Karaoke, music, radio, firecrackers |
| | Social events: soccer, wedding |
| Human voice | Food sellers with loudspeaker |
| | Normal conversation, guided tours |

maintenance. There are also guided tours for visitors and occasional social events. To account for the various sources of anthropogenic disturbance (Table 1), we subdivided the study area into three sectors characterized by high, moderate and low levels of human presence (Fig. 1B).

## Study animals

The six study groups consisted of a pair-bonded adult male and female with their putative offspring (4–6 individuals/group). Groups were labeled according to the distance between their home range and the highway. We used the northernmost limit of each home range as a measure of the groups' respective distance to the highway. Groups 1 and 2 resided in the cultivated garden where human presence culminated on a daily basis. Groups 3 and 4 resided mid-way across the park where human presence was moderate. Notably, group 3 occupied an area that overlapped an adjacent property on the west side of the park. Groups 5 and 6 resided in the native forest, which was least exposed to human presence. During a two-week-period preceding data collection, LHL set out to individually recognize the animals, based on their physical appearance, such as body size, age class, coat color pattern and facial characteristics. The putative offspring of all study groups consisted of an infant and juvenile, but some of the groups differed by the presence of a sub-adult (G2, G6) and un-weaned infants (G3, G4, G6; Table 2). To assess the age class of animals, we relied on the criteria established by other authors (*De Luna et al., 2010*; *Cäsar et al., 2012*).

## Data collection

### Acoustic survey methods

We used two different procedures to investigate the noise gradient in the study area. We began by measuring sound pressure levels (SPL) over a two-month period, and later we sampled the soundscape using passive acoustic monitoring (PAM) over a six-month period. While the first approach is based on direct measures of loudness, the second approach allows identifying which acoustic features best describe the noise gradient.

### SPL

Sound pressure was measured with a calibrated sound level meter (Voltcraft Plus SL-300), mounted on a tripod at ~1.50 m above ground. On the basis of similar studies (*Díaz, Parra & Gallardo, 2011*; *Duarte et al., 2011*), we elected the A-weighting filter, which

**Table 2 Information on group size and group composition.** Observation time for group movements and focal sampling is reported together with the number of tracks recorded, distance traveled by each group (mean ± SE) and traveling speed (mean ± SE).

| Group name | Group size | Group composition | Group movements | | | | | Focal sampling | |
|---|---|---|---|---|---|---|---|---|---|
| | | | D | H | T | Distance traveled (m) | Travel speed (m/h) | D | H |
| G1 | 4 | MFJI | 14 | 48 | 15 | 232.3 ± 36.9 | 91.4 ± 11.6 | 7 | 35 |
| G2 | 5 | MFSJI | 18 | 63 | 19 | 260.0 ± 49.9 | 95.3 ± 14.6 | 7 | 43 |
| G3 | 5 | MFJIB | 9 | 37 | 10 | 211.6 ± 53.2 | 81.9 ± 22.7 | 7 | 42 |
| G4 | 5 | MFJIB | 8 | 52 | 9 | 307.4 ± 57.8 | 63.2 ± 15.0 | 13 | 67 |
| G5 | 4 | MFJI | 9 | 42 | 9 | 269.3 ± 56.0 | 77.7 ± 12.1 | 9 | 39 |
| G6 | 6 | MFSJIB | 12 | 43 | 13 | 261.2 ± 54.0 | 80.3 ± 9.2 | 8 | 40 |

**Notes.**

D, number of days; H, number of hours; T, number of tracks; M, adult male; F, adult female; S, sub-adult; J, juvenile; I, infant; B, un-weaned infant.

approximates the sensitivity of the human ear (0.02–17.0 kHz). A-weighting, however, dampens low frequency noise, thus minimizing the true levels which would have been obtained with flat weighting (also called Z-weighting). The grid layout (180 × 800 m) consisted of three parallel transects roughly perpendicular to the highway and located on the east side, in the middle and on the west side of the park (Fig. 1C). Each transect comprised nine recording stations at ~100 m intervals thus resulting in 27 stations, all marked and geo-referenced (Garmin Etrex 30). At each station, we noted the maximum and minimum SPL values ('extreme values' hereafter) displayed on the instrument, after one minute of continuous recording with the sensor oriented towards the highway, and again with the sensor oriented towards Campo Verde. This was done to control for the directionality of the noise gradient. On a given day, the nine stations along a transect were sampled five times by one researcher (PA), at an interval of three hours starting at 6:00, 9:00, 12:00, 15:00 and 18:00 h. Three sampling days were necessary to cover the entire grid. This procedure was repeated on nine sampling days thus resulting in a total of 1,620 measures, with 15 pseudo-replications at each recording station (9 stations × 2 values × 2 orientations × 5 sampling hours × 9 sampling days).

### PAM

Passive acoustic monitoring was performed sequentially at 27 recording stations, all geo-referenced within the park (Fig. 2A). Soundscape data were acquired with a single H4n recorder (Zoom Corporation, Japan) connected to an Arduino UNO timer via the 5V input of the recorder. The ZOOM H4n was powered by a charger (Solar DC System XGX1206), which was connected to three 12V batteries mounted in parallel. Two photovoltaic solar panels were positioned 0–20 m from the recording device, in an area with good sun exposure to ensure that the batteries would operate overnight. All recording components were placed in a waterproof plastic box screwed to a metal mount, which was supported by two tripods at a height of 1.50 m. The microphone heads protruded from a small opening in the box and were protected by an anti-wind screen. The autonomous recording

Peer**J**

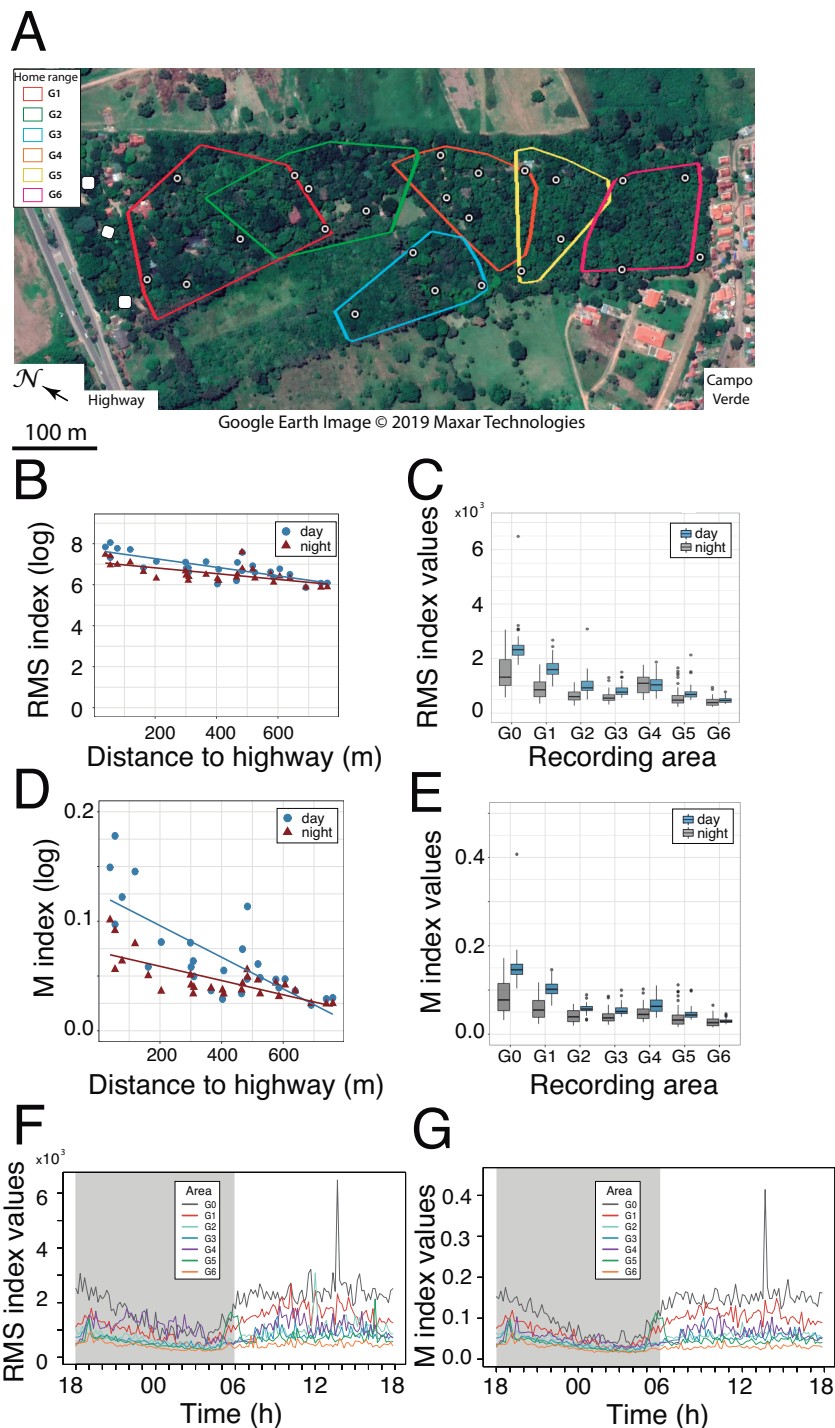

**Figure 2  Noise gradient and circadian variation in RMS and M acoustic indices.** (A) The 27 locations where passive acoustic monitoring was carried out. The autonomous recording unit was deployed at four sites within the home range of each study group. The three sites near the highway (white symbols) served as a baseline for anthropogenic noise (reference area G0). Map: Google Earth image ©2019 Maxar Technologies. (B and D) Linear regressions as a function of diel and distance to the highway shown for the RMS and M indices, respectively. (continued on next page...)

**Figure 2 (…continued)**
Each data point represents a mean value resulting from 72 one-minute samples at each recording site. (C and E) Boxplots comparing nocturnal and diurnal RMS values inside the titis' home ranges for the six studied groups (G1-G6) relative to the reference area G0. Note the non-linearity of the gradient due to increased RMS values in G4 home range. (E and F) Circadian variation in the RMS and M indices. Colored traces display the mean values from three sites near the highway (reference area G0) and from four sites inside each home range (G1–G6).

unit (ARU) was deployed at four locations within the home range of each study group, according to a rotation schedule over the entire study period, thus yielding 24 recording locations (4 locations × 6 groups). The three additional sites near the highway (Fig. 2A) served as a baseline for anthropophony (road traffic) in a reference area called G0.

Our rotation schedule consisted in four cycles of recording sessions. For each cycle, only one location per home range was sampled for ~48 h on average before moving to the next location, i.e., sequentially across groups. We rotated the recording unit every 5.0 ± 2.9 days (mean ± sd), which includes the time (>24h) necessary to recharge the batteries after a recording session. PAM started at 18:00 h with the recording level set to 100 (maximal sensibility). The ARU recorded one minute of ambient sound every 10 min for a minimum of 24 h. The sampling rate was 44,100 Hz with a 16-bit depth. The "mono mix" function of the Zoom H4n recorder was selected to mix the left and right channels. This recording schedule yielded 144 one-minute sound files per 24 h, which were saved in WAV format on a 32GB Secure Digital memory card.

Because the sound was identical on both channels, only the left channel was analyzed using the Seewave package (*Sueur, Aubin & Simonis, 2008*) available in the R environment (v.3.6.1, *R Core Team, 2019*). Moreover, given the variable duration of the recordings (range: 24–65 h), we analyzed only those samples collected during the first 24 h. For each sample, we computed the values of two acoustic indices, which were developed for rapid biodiversity assessment (*Sueur et al., 2014*): (1) the root-mean-square (RMS) index and (2) the median amplitude envelope (M) index. The RMS index computes the averaged dispersion of the amplitude values according to the formula:

$$RMS = \frac{\sqrt{x_1^2 + x_2^2 + \ldots + x_n^2}}{n}$$

where "$x_1 \ldots x_n$" corresponds to the amplitude values (positive or negative) of the signal in a given time interval and "$n$" the number of sound samples. This metric provides a measure of the intensity of the sound (*Rodriguez et al., 2013*; *Sueur et al., 2014*) and has been shown to be positively associated with increasing vocal activity along a gradient of habitat degradation (*Eldridge et al., 2018*). The M metric, another intensity index, is computed according to:

$$M = \text{median}(A(t)) \times 2^{(1-\text{depth})}$$

where A(t) is the amplitude envelope and depth the digital resolution of the signal (16 bits). The resulting values are then standardized between 0 and 1 by dividing each one by the maximum value. The M index was first used to estimate the number of animal vocalizations in a temperate woodland (*Depraetere et al., 2012*).

**Table 3 Categories of behavior distinguished during focal sampling.** In the text, the terms moving and traveling are used interchangeably. Similarly, we consider observing and vigilance as equivalent.

| Behavioral categories | Acronym | Description |
| --- | --- | --- |
| Resting | RE | Asleep (eyes closed, head down) |
| | | Awake and remains still |
| | | Basking on the edge or on top of the canopy |
| Observing | OB | Scanning the environment with noticeable head movements while remaining stationary on a branch |
| Moving | MO | Quadrupedal locomotion, climbing, leaping |
| Foraging | FO | Searching for and consuming food items (fruits, flowers, leaves and insects) |
| | | Grooming |
| | | Agonistic interaction |
| Socializing | SO | Play |
| | | Tail twining |
| | | Vocalizing loudly or softly |

## Behavioral observations

From September 26, 2017 till January 16, 2018, activity budgets of family groups were evaluated from 51 days of focal sampling (266 h total). On the day preceding data acquisition, the observer (LHL) located and followed the focal group until it elected its sleeping tree. The focal group was followed again from that location next day. In the case the observer failed to identify the sleeping tree the day before, then the first morning vocalizations served to locate the focal group. A "session of observations" started from the moment the focal group was found until it was lost. In this case, a new search began until the focal group was found again, thus allowing a second "session of observations".

Groups were observed throughout the day, according to a pre-established schedule consisting in acquiring data sequentially from each group on a monthly basis. Each individual in the group was monitored for half an hour, using the focal-animal sampling method (*Altmann, 1974*). That is, every five minutes, the focal individual was observed for one minute, its activity being assigned to one of five main activities: resting (RE), observing (OB), moving (MO), foraging (FO), and socializing (SO) (for more details, see Table 3). The observer also estimated the distance separating the focal individual to the nearest group member and the location of the focal individual in each of three forest strata (S1: 0–5 m, S2: 5–10 m; S3: 10–15 m). Pentax binoculars, 10 × 50 magnification, were used to observe the titis when animals were distant.

The paths of each family group were mapped during focal observations on a satellite image of the study area, which contained numerous landmarks and a network of geo-referenced trails. Subsequently, the tracings were reconstructed in Google Earth Pro and the size of each home range was determined according to the method of the minimum convex polygon (*Odum & Kuenzler, 1955*). Group travel speed was computed for each path by dividing the distance traveled by the duration of a session of observations.

This observation schedule yielded a total of 2,897 behavioral events from which activity budgets were computed. The number of events collected for each group was as follows: G1

($n = 391$), G2 ($n = 503$), G3 ($n = 456$), G4 ($n = 735$), G5 ($n = 446$), and G6 ($n = 366$). For each group member, we calculated the number of times this individual was engaged in a given activity and the respective frequencies for that activity (calculated by adding up the number of times spent in an activity and then dividing by the sum of all activities collected within the group). These numbers are reported as proportions of time spent by the group in each activity. Similarly, the relative number of records in which the focal individual was observed higher or lower than 5 m above ground gave us the proportion of time spent by group members in each forest stratum.

As mentioned above, we collected data of proximity between a focal individual and its nearest companion. We call such transient association a "social dyad". Note that the number of neighbors can vary from 0 (absence of neighbor referred to as "solo") up to the size of the whole group ("grouping"). To quantify the degree of association between group members, we calculated the respective proportions of solos, social dyads and groupings. For social dyads and groupings, the data were split into two categories. "Tight social dyads" were those where the nearest neighbor was located within a radius of 1 m from the focal individual and "loose social dyads" were those where the nearest neighbor was located within a radius of 1.5 to 7 m from the focal individual. We made a similar distinction between "tight groupings" and "loose groupings". We chose this threshold because in each group the vast majority of dyads and groupings were recorded within a 1m radius.

### Field experiments

To investigate whether titi monkeys respond to human presence as they do when facing a potential predator, an experiment was carried out where monkeys were exposed to a clothed mannequin sitting on a chair (Fig. 3). On the day prior to an experimental session, one observer (LHL) followed the experimental group until it settled for the night. Next morning, before dawn, the mannequin was placed in an open area, where it was predicted that the group would emerge after leaving the sleeping tree. The two observers waited for the titis to arrive while concealed in the vegetation, 5–10 m away from the mannequin. Upon the titis' arrival, one observer (LHL) reported at low voice the *ad libitum* behavior of group members on a mobile phone while a second observer (PA) videotaped the whole episode with a Canon XL-H1 camcorder equipped with a wide-angle lens and a Seenheiser ME66-K6P microphone.

To prevent the monkeys from habituating to the mannequin, we performed only one successful session per group. A "successful session" was an experimental session in which at least one individual in the group saw the mannequin. If the group failed to appear, the experimental session was reported as a "failed session". In this case, another experimental session was performed on a subsequent day until we got a successful session. An experimental session ended when all group members withdrew from the area. The six groups of titi monkeys were partly habituated to the presence of one observer (LHL) when we began testing. It is unlikely that the titis responded to the concealed observers as they seemed more distracted by the mannequin, which was quite visible on the ground.

The video recordings were visualized with Adobe Premiere Pro CC (v. 2015). Despite having used a wide-angle lens for filming the titis, not all group members were visible in

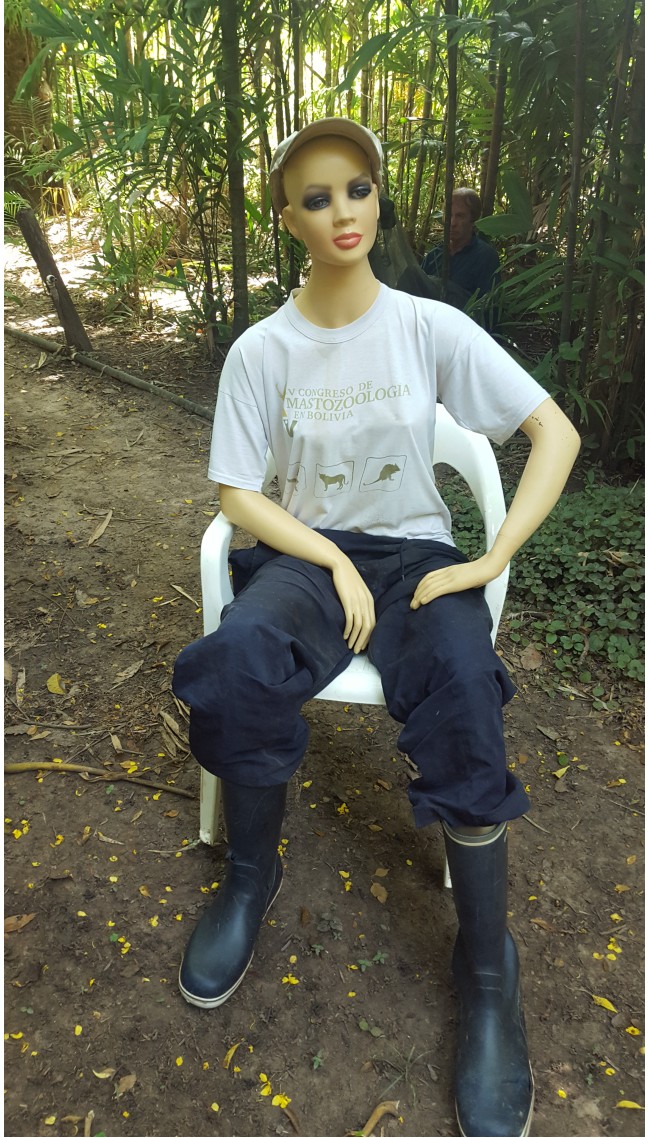

**Figure 3** **Mannequin used in field experiment.** One observer is concealed in the background. Photo credit: Lucero M. Hernani Lineros.

the video clips. For a detailed description of the titis' behavior during a successful trial, we mostly relied on the *in-situ* observations (Article S1). Nevertheless, the video footage was useful to confirm which group member approached the mannequin and estimate the approach distance. The titis' vocal response was highly variable between groups and, in contrast with the conspicuous mouth movements associated with loud calling, the soft and high-pitched alarm calls are produced with the mouth barely opened. Thus, in most trials, we were unable to rigorously identify the callers, both *in situ* and from the video footage. We therefore exported the sound track as a WAV file and generated spectrograms of the calls using Raven Pro (v. 1.5). For scoring, we followed Cäsar's classification scheme of

black-fronted titis, *Callicebus nigrifrons*, alarm calls, based on call shape (*Cäsar et al., 2012*). We carefully inspected the spectrograms and counted all occurrences of each call type (A 'chirp', B 'cheep', C 'squeaks', W 'whistle'; Fig. S1) emitted "per group" during a given trial. Except for the whistle, these faint, high-pitched alarm calls, less than 100 ms in duration, were delivered in short series at a rate of 6 to 8 calls/s. All exhibited a putative fundamental sound (F0) between 4 and 5 kHz with at least one visible harmonic. To standardize the data, we retained those emitted in the first minute following detection of the mannequin by the group (*Cäsar et al., 2013*).

## Fecal sampling and cortisol enzyme immunoassays (EIA)

During behavioral observations, the observer paid particular attention to fecal deposition by resting group members. When defecation occurred, the site where the droppings landed was immediately searched and the fresh sample was collected with disposable gloves. Such diversion terminated a session of observation and usually resulted in the loss of visual contact with the group. From each sample, 0.1 g of stool was extracted, weighed with a portable precision scale (OHAUS model YA302) and then placed in 15 ml polypropylene tubes containing a mix of 2.5 ml of distilled water and 2.5 ml of ethanol (*Ziegler & Wittwer, 2005*). Each tube was tightly capped, sealed with parafilm and carefully labeled with a date, sample number, individual identity and time of fecal deposition. We set out to collect as many fresh samples as possible from each individual but ended up with fewer than we thought, which is a weakness of our study. The 30 fecal samples were collected opportunistically throughout the day (Table 4), and came from 19 identified individuals (out of the 26 monitored individuals). The samples were stored in a freezer until they were exported and processed for cortisol analysis at the Paleogenomic and Molecular Genetics laboratory at the Musée de l'Homme in Paris, France. Using the salivary cortisol assay kit from Salimetrics ©, enzyme-immunoassays were performed to assess the effectiveness of cortisol antibodies in binding to fecal metabolites in Bolivian gray titi monkeys. We provide a technical report of parallelism and validation tests of this glucocorticoid (Article S2). All applicable institutional, national and international guidelines for the care and use of animals were followed. We obtained research approvals and export permits from the National Biodiversity Authority in Bolivia (MMAYA-VMABCCGDF-DGBAP/MEG N°0531/2018) and from the National Service of Food Safety and Agricultural Health (SENASAG 0040452), respectively, and import permit from the Direction Départementale de la Protection des Populations de Paris (N°2018-75-31555).

## Statistics

To account for temporal pseudoreplication, sound pressure values were fitted to a generalized mixed-effects model in which three categorical variables (orientation, extreme values, time of day) and one covariate (distance to highway) were entered as fixed effects. Because measures obtained from each transect were made on separate days, we included a random effect term for varying intercepts by transects, and for recording stations that are nested within transects. To examine the changes occurring in both acoustic indices as a function of distance to the highway, we performed Least Squares Regression analyses on the log transformed data. For normality assumptions, we relied on the Shapiro–Wilks test.

**Table 4 Cortisol concentrations.** The 30 fecal samples have been sorted by group, individual and age category.

| Group | Sample # | Date (MMDDYY) | Individual | Age category | Time of day | Cortisol cc. ($\mu$g/dL) |
|---|---|---|---|---|---|---|
| G1 | 15 | 12/09/17 | F | A | 07:22 | 0.142 |
| G1 | 29 | 04/05/18 | F | A | 10:26 | 0.182 |
| G1 | 30 | 04/05/18 | F | A | 10:26 | 0.175 |
| G1 | 16 | 12/09/17 | J | Y | 07:22 | 0.110 |
| G1 | 17 | 12/09/17 | J | Y | 07:22 | 0.155 |
| G2 | 9 | 11/22/17 | F | A | 16:56 | 0.071 |
| G2 | 18 | 12/12/17 | M | A | – | 0.207 |
| G2 | 19 | 12/12/17 | M | A | – | 0.113 |
| G2 | 10 | 11/23/17 | S | A | 14:40 | 0.092 |
| G2 | 21 | 01/16/18 | J | Y | – | 0.140 |
| G2 | 28 | 04/05/18 | J | Y | 10:26 | 0.051 |
| G3 | 11 | 11/28/17 | F | A | 15:24 | 0.108 |
| G3 | 12 | 11/28/17 | F | A | 15:24 | 0.095 |
| G3 | 23 | 03/20/18 | M | A | 10:29 | 0.105 |
| G3 | 24 | 03/20/18 | M | A | 10:29 | 0.089 |
| G3 | 6 | 11/11/17 | J | Y | 09:47 | 0.114 |
| G3 | 5 | 11/11/17 | I | Y | 09:47 | 0.116 |
| G4 | 7 | 11/13/17 | F | A | 17:42 | 0.141 |
| G4 | 25 | 03/22/18 | M | A | 09:41 | 0.115 |
| G4 | 20 | 12/19/17 | J | Y | 13:55 | 0.095 |
| G5 | 1 | 10/17/17 | F | A | 18:22 | 0.319* |
| G5 | 13 | 12/02/17 | F | A | 10:29 | 0.344* |
| G5 | 22 | 03/16/18 | F | A | – | 0.160 |
| G5 | 26 | 03/28/18 | F | A | 10:33 | 0.179 |
| G5 | 2 | 10/30/17 | M | A | 15:51 | 0.103 |
| G5 | 14 | 12/02/17 | J | Y | 10:29 | 0.092 |
| G5 | 27 | 03/28/18 | J | Y | 10:33 | 0.163 |
| G6 | 8 | 11/18/17 | S | A | 12:11 | 0.103 |
| G6 | 4 | 11/03/17 | J | Y | 17:01 | 0.150 |
| G6 | 3 | 11/03/17 | I | Y | 17:01 | 0.100 |

**Notes.**

A, adult; Y, young; F, female; M, male; S, subadult; J, juvenile; I, infant.

Asterisks denote the two highest values, which were measured in one adult female from G5.

To assess whether titis' behavior was affected by the disturbance (anthropophony and anthropic activity), we applied the Chi-square test goodness of fit to compare within-group frequencies of behavior and the number of individual occurrences in each forest level. The Pearson's Chi-square was performed to compare the between-group activity budgets and forest level utilization. We did not perform the test when the cells in the contingency tables contained an expected frequency less than 1 or less than 5 in more than 20% of cases. Second, to analyze changes in activity budgets as a function of distance to the highway, we performed analyses of covariance, following *Crawley's (2014)* recommendations. Here, the

response variable was a matched pair of counts (occurrences in one behavioral category vs. other categories) that we wished to analyze as proportion data. Therefore, we applied generalized linear models (GLM) fitted for binomial data in which two categorical variables (sex, age) and one covariate (distance to highway) were entered as fixed effects.

To assess whether titis responded differentially to the mannequin in each of the three areas of varying anthropic disturbance, we used a G test for goodness of fit to determine if there was an overall difference in the proportion of calls emitted per group and per area of human disturbance.

To account for the variation in the number of individual fecal samples within each group, we performed a mixed-effects linear model in which we evaluated the effect of group distance (as covariate) on cortisol concentration, with a random factor in which individuals were nested within group. In addition, age was entered in the model as a categorical fixed factor by aggregating group members into either 'adult' category, which included adult males, adult females and sub-adults ($n = 11$) or 'young' category, which included juveniles and infants ($n = 8$). To relate cortisol concentrations with each class of noise level, we used a Pearson correlation test. We first calculated the mean RMS and M values obtained at each site in the course of 24 h. These mean values were then averaged across the four sampling sites of each home range, which gave us six overall mean values for each acoustic index. Similarly, given little within-group variation (no sex or age difference), cortisol concentrations were averaged, thus resulting in six mean cortisol values that were representative of each group.

The models were fitted with restricted maximum likelihood, using the 'nlme' R package (v.3.6.1) and we selected the one with the lowest Akaike information criterion (AIC). Pearson residuals were visualized with the package 'corrplot' of R and post-hoc tests were performed with Bonferroni corrections. All statistical models are reported in Table S1.

## RESULTS

### Sound pressure gradient

GLMM analysis using the log transformed data pointed to the model devoid of interactive factors as the most parsimonious one, based on Akaike information criterion (Table S1.1). The additive model detected significant contributions of each factor (Wald test: $\chi^2_{orientation} = 5.75$, $df = 1$, $p = 0.016$; $\chi^2_{extreme\ values} = 1698.13$, $df = 1$, $p < 0.001$; $\chi^2_{time\ of\ day} = 90.59$, $df = 4$, $p < 0.001$) and the covariate (Wald test: $\chi^2_{distance} = 41.52$, $df = 1$, $p < 0.001$) on sound pressure (Fig. 4A). Irrespective of the sensor orientation (northward or southward), there was a non-linear decay process in sound pressure from the highway towards Campo Verde (Fig. 4B). Moreover, minimum SPL fitted the regression line better than maximum values ($var_{minN} = 66.3$; $var_{minS} = 61.0$; $var_{maxN} = 152.9$; $var_{maxS} = 140.0$). Overall, minimum SPL dropped sharply from the highway (mean ± SD: $66.4 \pm 3.0$ dB) to a distance of 100 m ($52.5 \pm 4.5$ dB) with an additional decay of 8.3 dB up to 800 m ($44.2 \pm 4.4$ dB). Linearity in the dataset was obtained after excluding SPL measurements at 1m from the highway. Subsequently, ANCOVA performed on the restricted dataset (100–800 m) revealed a significant interaction between distance and

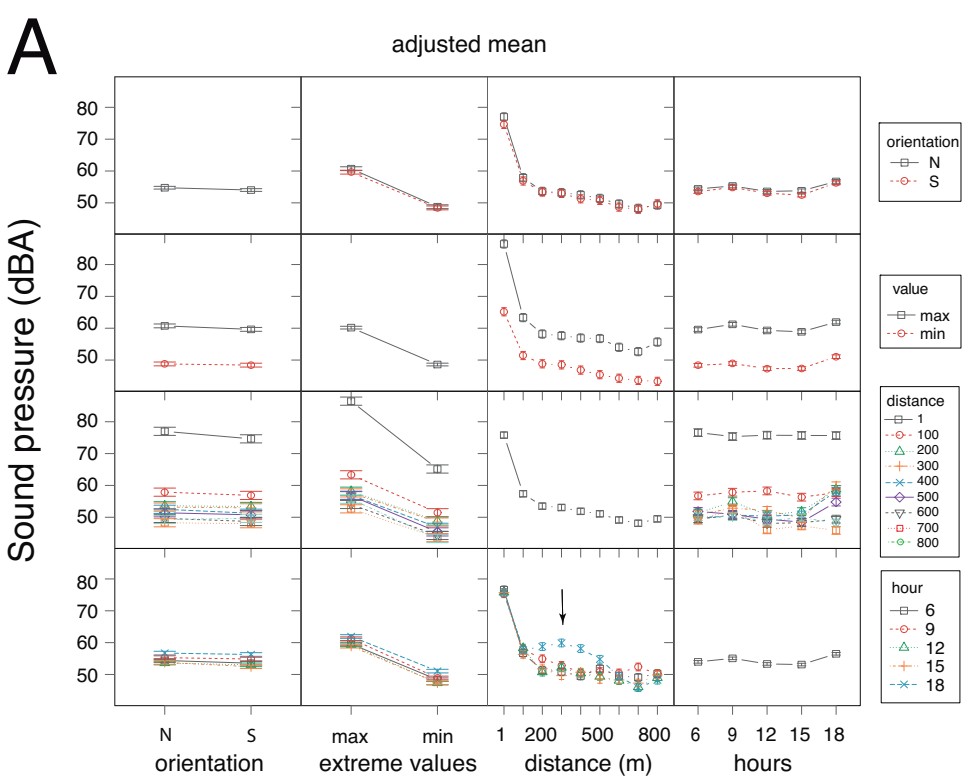

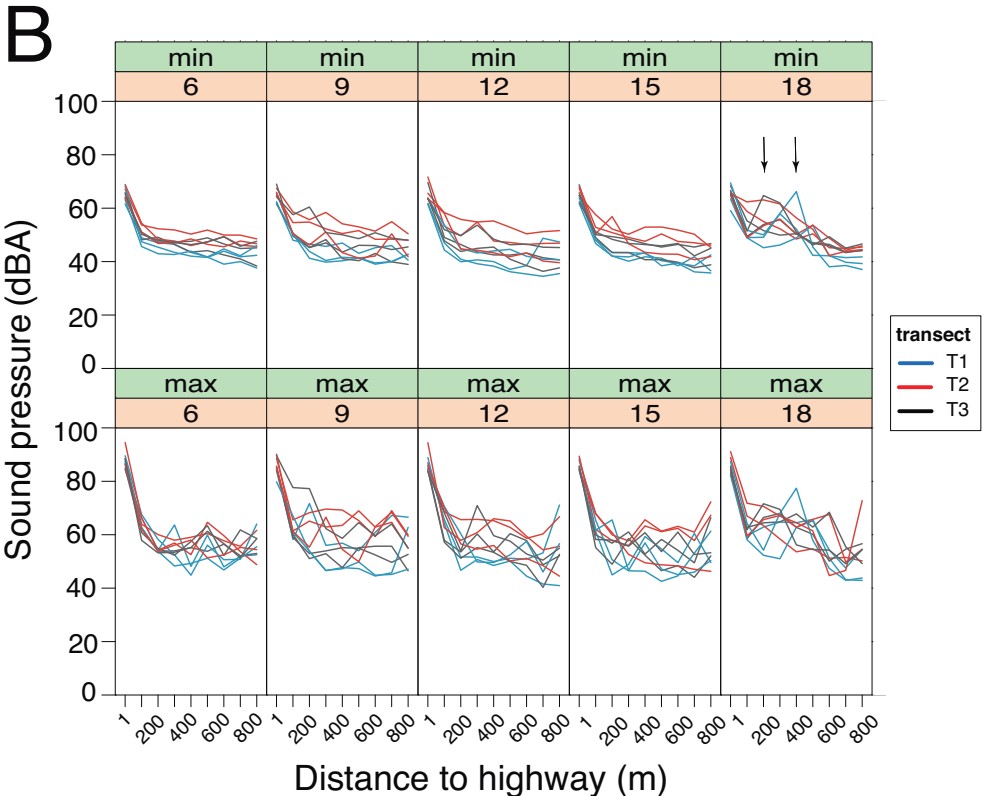

**Figure 4** **Sound pressure gradient.** (A) Interaction plots generated by the linear mixed model. (continued on next page...)

**Figure 4 (...continued)**
The fixed variables that were entered in the full model are shown at bottom. In the evening hours (18:00 h), the cicadas' chorus produced a significant increase in sound pressure levels (vertical arrow). N, North orientation; S, South orientation; min, minimal values; max, maximal values. (B) Changes in sound pressure levels as a function of distance to the highway and time of day. Minimal (min) and maximal (max) values are plotted separately. Within each panel, the values obtained for each sampling day are connected by lines, which are color-coded per transect (T1: blue, T2: red, T3: black). The non-linear noise gradient is more obvious for minimum values than it is for maximum values. Vertical arrows point to an increase in the minimum sound pressure levels resulting from the cicada's chorus at dusk.

time-of-day (h18 $\times$ distance: $t = -2.919$, $p = 0.004$; Table S1.2). Specifically, we observed little change in the noise gradient during the course of the day, except at 18:00 h when sound pressure levels increased significantly due to the onset of the cicadas' dusk chorus (Fig. 4).

## Characterization of noise

Both RMS and M acoustic indices were inversely correlated with the distance to the highway, exhibiting a steeper slope during the day, as compared with nocturnal values (Pearson correlation test for RMS: $r_{day} = 0.78$, $p < 0.001$, $r_{night} = 0.67$, $p < 0.001$; Pearson correlation test for M: $r_{day} = 0.80$, $p < 0.001$, $r_{night} = 0.78$, $p < 0.001$; Figs. 2B and 2D). Inside each group's home range, except for RMS values in G4 (Fig. 2C) and M values in G6 (Fig. 2E), both indices were significantly higher during the day than during the night (paired $t$-tests: $t \geq 3.13$, $df = 71$, $p < 0.01$; $t \geq 4.52$, $df = 71$, $p < 0.001$, respectively). Furthermore, both indices exhibited a strong circadian pattern in the reference area (G0), with a gradual attenuation at increasing distances from the highway (Figs. 2F and 2G).

## Time budgets, social relationships and use of space by the titis

First, we examined whether home range size differed among family groups. The reconstruction in Google Earth of 75 tracks (mean length $\pm$ SD: 255.0 $\pm$ 178.9 m) obtained from 282 h of behavioral observations revealed small, overlapping home ranges (range: 1.35–3.38 ha) with groups G1 and G2 exploiting a relatively larger area (Figs. 2A and 5A). We found no difference in the groups' travel speed after log-transformation of the data (mean $\pm$ SD: 81.6 $\pm$ 11.3 m/h, ANOVA: $F_{5,69} = 0.84$, $p = 0.52$; Table 2). Daily paths obtained from three complete days in three groups averaged 569.3 $\pm$ 36.9 m.

Second, we examined whether activity budgets and use of forest strata differed among family groups. Within each group, the titis spent most of their time resting (mean $\pm$ SD: 38.6 $\pm$ 6.3%) and allocated less time to moving (22.8 $\pm$ 2.7%), foraging (21.1 $\pm$ 4.3%), observing (11.6 $\pm$ 3.7%) and socializing (5.9 $\pm$ 2.3%; Figs. 5B and 5D). When socializing, the main behaviors were vocalizing (47.3 $\pm$ 20.2%, $n = 70$), grooming (22.6 $\pm$ 14.8%, $n = 40$) and tail twining (15.2 $\pm$ 14.3%, $n = 30$) whereas both play and agonistic behaviors occurred sporadically (8.3 $\pm$ 6.3%, $n = 15$ and 6.6 $\pm$ 5.3%, $n = 9$, respectively). Interestingly, family groups spent significantly more time in the lower forest stratum than higher up (S1: 68.74 $\pm$ 7.92%, S2: 28.82 $\pm$ 7.49%, S3: 2.43 $\pm$ 1.94%; Wilcoxon signed-rank test comparing S1 vs. S2+3: $V = 0$, $p = 0.031$; Figs. 5C and 5E). Moreover, their activity budgets differed accordingly (Table 5): moving occurred more frequently than expected in

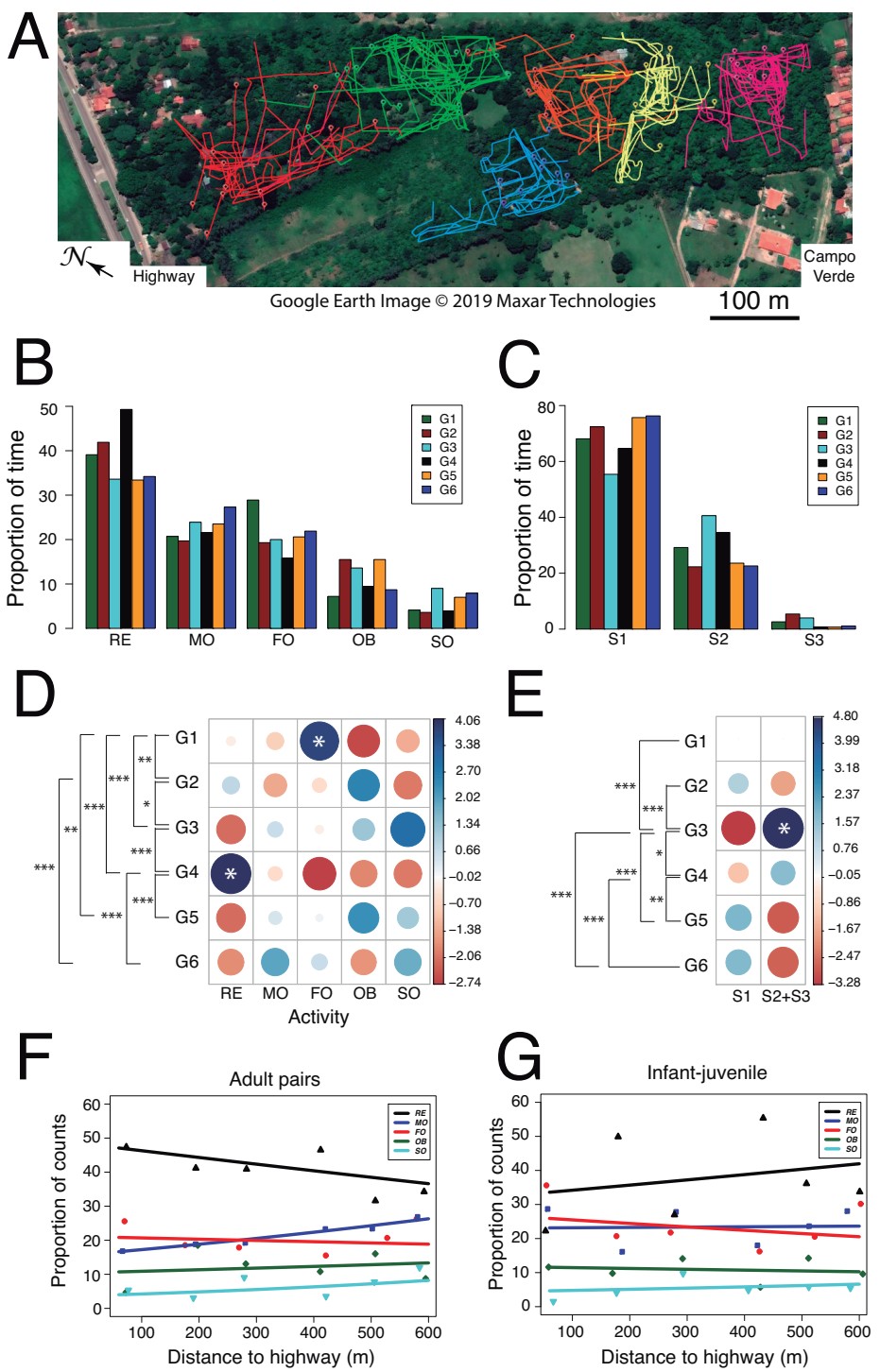

**Figure 5** **Use of space and activity-budget.** (A) Reconstruction of the 75 travel paths registered during the study period. A color-code distinguishes between family groups and symbols denote the starting point of a given path. Map: Google Earth image © 2019 Maxar Technologies.(continued on next page…)

**Figure 5 (…continued)**
(B) Histograms of group activity budgets: proportion of time spent by each group of titis performing each activity (RE: resting, MO: moving, FO: foraging, OB: observing, SO: socializing). (C) Histograms of utilization of forest strata: proportion of time spent by the titis in each forest level (S1: 0–5 m, S2: 5–10 m; S3: 10–15 m). For quantitative analysis, data from S2 and S3 were lumped, owing to fewer occurrences in S3. (D) Graphical output of standardized residuals for group activity budgets. (E) Graphical output of standardized residuals for utilization of forest strata. (F) Activity budgets of adult pairs and (G) infant-juvenile pairs are plotted as a function of the nearest distance of their home range to the highway.

the lower stratum (S1: $26.01 \pm 1.98\%$ vs. S2+S3: $13.82 \pm 3.97\%$; Wilcoxon signed-rank test: $V = 0$, $p = 0.031$) whereas foraging tended to be more frequent than expected in the higher strata (S1: $18.07 \pm 4.87\%$ vs. S2+S3: $29.50 \pm 7.84\%$; Wilcoxon signed-rank test: $V = 1$, $p = 0.063$). Analysis of standard residuals revealed significant between-group differences both in activity budgets (Pearson's Chi-square: $\chi^2 = 107.4$, $df = 20$, $p < 0.001$) and use of forest strata ($\chi^2 = 64.5$, $df = 5$, $p < 0.001$). ANCOVA analysis per age class (adult vs. young) disclosed subtle but significant changes in specific activities according to home range distance to the highway (Table S1.3). In particular, regression analysis with binomial errors showed that the time spent moving by adult pairs increased significantly with home range distance to the highway (male–female: $z_{distance} = 3.06$, $p = 0.002$; $z_{sex} = 0.74$, $p = 0.461$; Fig. 5F) and this tended to be significant for each sex taken separately ($z_{female} = 2.53$, $p = 0.011$; $z_{male} = -1.85$, $p = 0.065$). No such relationship was found in the activity budget of young animals (infant-juvenile: $z_{distance} = 0.30$, $p = 0.77$; $z_{age} = -1.67$, $p = 0.08$; Fig. 5G). Thus, adult pair locomotion was inversely related to the gradient observed in both acoustic indices (RMS and M).

Lastly, we examined whether social relationships differed among family groups. From a total of 2439 measures of social proximity, tight social dyads (63.9%) were more frequent than loose social dyads (13.7%). Likewise, tight groupings (18.5%) occurred more frequently than loose groupings (0.5%) and the focal animal was rarely found solo (3.5%). Tight social dyads ($n = 1558$) consisted predominantly of the adult pair (FM: $31.0 \pm 10.3\%$), less frequently of one adult next to a juvenile (FJ: $14.8 \pm 6.5\%$, MJ: $10.5 \pm 6.3\%$) or next to an infant (FI: $13.9 \pm 6.7\%$, MI: $13.5 \pm 4.4\%$). Still fewer records entailed a juvenile next to an infant ($8.6 \pm 6.0\%$) whereas social dyads involving a sub-adult were least observed (SF: $9.0 \pm 0.1\%$, SM: $4.1 \pm 5.8\%$, SJ: $4.6 \pm 1.4\%$, SI: $5.1 \pm 4.0\%$). Overall, we found no evidence of a gradual change in social relationships as a function of group location relative to the highway.

## Response to the mannequin

In total we performed 18 experimental sessions, 12 of which were unsuccessful (66.7%). For each successful session ($n = 6$), group members approached the visual model within a radius of 4 m with a mean latency of $86.7 \pm 16.9$ min (Table 6). As we expected, the two groups most exposed to human disturbance showed either no obvious reaction to the mannequin (G1) or reacted weakly with piloerection and brief calling (G2). By contrast, groups residing in quieter areas reacted either strongly (G3: 150 calls during the first minute) or moderately (G4: 62 calls, G5: 59 calls, G6: 71 calls) (G test: $G_{group} = 325.87$; $df = 5$, $p < 0.001$; $G_{area} = 286.4$; $df = 2$, $p < 0.001$). While looking at the mannequin, the

**Table 5  Family group activity budgets.** The proportion of time spent by each group (G1–G6) in distinct categories of activity is reported separately for each forest stratum (height) after pooling data from strata 2 and 3 ( >5).

| Activity | Height (m) | G1 | G2 | G3 | G4 | G5 | G6 | Mean ± SD % | Wilcoxon p-value |
|---|---|---|---|---|---|---|---|---|---|
| RE | ≤ 5 | 39.5 | 36.8 | 34.7 | 55.1 | 34.5 | 37.8 | 39.73 ± 7.76 | $V = 9$ |
|  | >5 | 41.0 | 55.4 | 32.7 | 38.5 | 29.9 | 38.1 | 39.27 ± 8.90 | $p = 0.844$ |
| OB | ≤ 5 | 7.7 | 18.1 | 8.0 | 5.1 | 14.7 | 11.1 | 10.78 ± 4.87 | $V = 8$ |
|  | >5 | 6.6 | 8.6 | 20.8 | 17.5 | 18.7 | 2.4 | 12.43 ± 7.54 | $p = 0.688$ |
| MO | ≤ 5 | 24.9 | 24.2 | 28.3 | 24.1 | 26.1 | 28.5 | 26.01 ± 1.98 | $V = 0$ |
|  | >5 | 10.7 | 7.9 | 17.3 | 16.0 | 13.1 | 17.9 | 13.82 ± 3.97 | $p = 0.031$ |
| FO | ≤ 5 | 24.9 | 17.3 | 21.5 | 10.5 | 16.8 | 17.4 | 18.07 ± 4.87 | $V = 1$ |
|  | >5 | 35.2 | 24.5 | 18.3 | 26.1 | 33.6 | 39.3 | 29.50 ± 7.84 | $p = 0.063$ |
| SO | ≤ 5 | 3.1 | 3.6 | 7.6 | 5.1 | 7.8 | 5.2 | 5.40 ± 1.96 | $V = p =$ |
|  | >5 | 6.6 | 3.6 | 10.9 | 1.9 | 4.7 | 2.4 | 5.02 ± 3.34 | 1.0 |

**Notes.**
RE, resting; OB, observing; MO, moving; FO, foraging; SO, socializing.

**Table 6  Summary of field experiments with the mannequin.** Tests were performed in each of three areas exhibiting different levels of human presence. For each group, the table reports the total number of trials performed to achieve a successful test, characterized by the following parameters: wait period, individual who first detected the mannequin, approach distance to the mannequin, calling duration, number of calls emitted during first minute and call type.

| Area | Human presence | Group name | # trials | Wait period (min) | Indiv. | Approach distance (m) | Calling duration (s) | # calls (1st min) | Call type |
|---|---|---|---|---|---|---|---|---|---|
| 1 | High | G1 | 1/6 | 120 | M | 3.0 | – | 0 | – |
|  |  | G2 | 1/3 | 25 | I | 1.5 | 3 | 1 | – |
| 2 | Moderate | G3 | 1/2 | 59 | F | 4.0 | 180 | 150 | B |
|  |  | G4 | 1/1 | 126 | I | 3.0 | 148 | 54 - 8 | B - C |
| 3 | Low | G5 | 1/4 | 120 | F | 1.5 | 90 | 58 - 1 | B - W |
|  |  | G6 | 1/1 | 70 | – | 4.0 | 131 | 71 | B |

**Notes.**
M, male; F, female; I, infant; B, call B; C, call C; W, whistle.

titis from these groups exhibited typical postures with signs of arousal, such as arching the back with piloerection (Article S1). From a total of 343 vocal emissions, three types of calls were identified (Table 6): the most common type was call B (97.4%), then call C (2.3%), lastly the whistle (0.3%). However, against our prediction, the titis' alarm response to the mannequin did not squarely correlate with group location relative to the highway as groups G5 and G6 responded less to the mannequin than did groups G3 and G4 (post-hoc tests: $\chi^2_{\text{area1}} = -12.98$, $p < 0.001$, $\chi^2_{\text{area2}} = 11.19$, $p < 0.001$, $\chi^2_{\text{area3}} = 1.79$, NS).

## Fecal cortisol concentrations

Enzyme immunoassays of 30 fecal samples issued from 19 individuals detected low cortisol concentrations (mean ± SD: 0.14 ± 0.06 µg/dL, range: 0.05–0.34 µg/dL) with little between-group variation (Table 4). Two samples were three standard deviations above average but the normality assumption was met after log-transformation of the data (Shapiro–Wilks test: $W = 0.95$; $p = 0.16$). A linear mixed effects analysis with age and

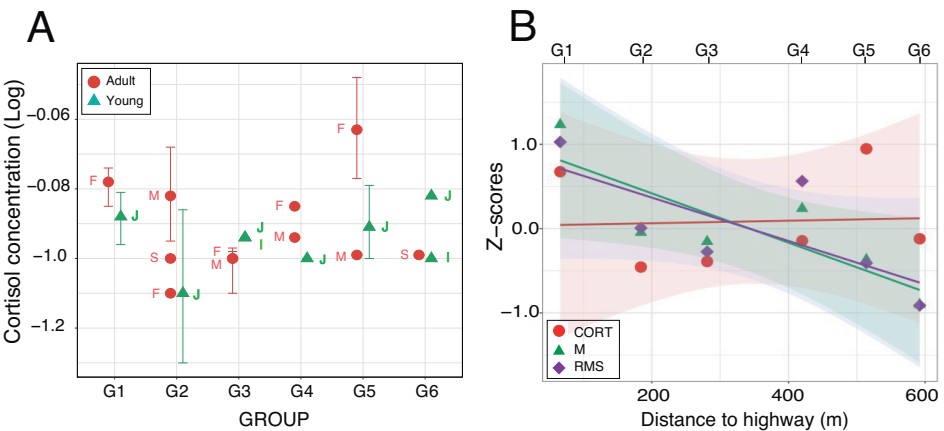

**Figure 6 Fecal cortisol concentrations.** (A) Mean values (±95% CI) plotted for each of the nineteen individuals from which one fecal sample at least was obtained. Labels identify the individual age-sex classes: I, infant; J, juvenile; S, sub-adult; M, adult male and F, adult female. (B) Relationships between cortisol concentrations and acoustic indices after $z$-score transformation. For each group, mean values have been plotted as a function of the nearest distance of their home range to the highway. Regression lines are displayed with their respective 95% confidence intervals.

distance as fixed factors and individuals nested within group as a random factor revealed no significant difference between the full and reduced models (Table S1.4). The retained model detected no significant effect of the fixed factors (Wald test: $\chi^2_{age} = 0.84$, $df = 1$, $p = 0.36$; $\chi^2_{distance} = 0.25$, $df = 1$, $p = 0.62$). The random effect accounted for 48.4% of the variance left over after the variance explained by the fixed effects (Fig. 6A). The timing of sample collection had no effect on cortisol concentration (morning: $0.14 \pm 0.07$ µg/dL, $n = 15$; afternoon: $0.12 \pm 0.07$ µg/dL, $n = 11$; Wilcoxon rank sum test: $W = 53$, $p = 0.13$). Pearson regression analysis found both acoustic indices highly correlated (M-RMS: $F_{1,4} = 56.91$; $p = 0.002$). Against our prediction, however, we found no relationship between hormonal concentrations and the log of the noise gradient in the study area (M-CORT: $F_{1,4} = 0.26$; $p = 0.635$; RMS-CORT: $F_{1,4} = 0.12$; $p = 0.744$; Fig. 6B).

## DISCUSSION

In line with our hypothesis, the anthropogenic noise seemed to impact the titis' behavior in two ways. First, as a between-group variation in the activity budgets of adult pairs: the more exposed they were to anthropophony, the less time they spent moving. Second, our field experiment using a mannequin indicated a reduction in the titis' alarm response as a result of habituation to human presence. Against our prediction, however, we found no evidence of stress-related elevation of fecal cortisol, as would be expected in animals exposed to chronic anthropogenic perturbation (*Cross, Pines & Rogers, 2004*; *Kaplan, Pines & Rogers, 2012*).

### Highlighting the anthropogenic noise gradient in the titis' habitat

We confirmed the presence of a noise gradient in the study area. First, sound pressure measurements with the sound level meter oriented northward and then southward clearly

point to noise attenuation from the highway up to the suburb of Campo Verde, and not the reverse. This shows that roadway traffic, not the suburb, was the main source of the noise gradient. Of interest, maximal sound pressure values exhibited more variability than did minimal sound pressure values. The noise gradient persisted throughout the day with little fluctuation, except in the evening hours with the onset of the cicadas' chorus. Indeed, cicadas may produce some of the loudest sounds among living organisms, as has been reported for *Cistosoma saundersii* with sound pressure peaking at 158 dB inside the air sac (*Young & Bennet-Clark, 1995*). Because of its persistence, the cicadas' dusk chorus affected both maximum and minimum SPL readings. Although less intense than stridulating cicadas, roadway traffic noise was perceptible from dawn to dusk, and into the night.

The native forest in the park might have acted as a noise barrier, especially for high-frequency sounds, but we found no clear pattern of excess attenuation. From 100 m off the highway up to the quiet suburb of Campo Verde, sound pressure decayed linearly and smoothly, with an 8.3 dB drop in background noise. Under free field conditions, a 6 dB drop is expected with a doubling of the distance from the sound source (inverse square law). Absence of excess attenuation can be attributed to the presence of multiple, heterogeneous sound sources, both spatially and temporally distributed in the study area. Although well-controlled sound transmission experiments might have confirmed a barrier effect of the native forest, this topic was outside the scope of our study.

Second, using passive acoustic monitoring, we confirmed the presence of a noise gradient, as evidenced by the RMS and M acoustic indices (*Depraetere et al., 2012*; *Rodriguez et al., 2013*; *Eldridge et al., 2018*). We were unable to dissociate the relative contribution of roadway traffic and human activity from the audio recordings. However, we note that the noise gradient persisted at night when human presence inside the park was highly reduced. Furthermore, compared with the nocturnal values of the acoustic indices, the diurnal values were generally higher and the noise gradient exhibited a steeper slope, supporting the view of increased diurnal anthropophony, roadway traffic and human activity combined (for similar circadian patterns in soundscape dynamics, see *Duarte et al., 2015*). Obviously, multiple recorders deployed in the titis' home ranges would have provided a more robust methodology to assess the moment-to-moment changes in soundscape across the entire area. Nevertheless, our rotation schedule with a single recording platform shows that relevant patterns of soundscape can be revealed over time with minimal equipment (see also *Turner, Fischer & Tzanopoulos, 2018*).

An important issue, of course, is whether titi monkeys are sensitive to traffic noise. Behavioral audiograms for callicebine monkeys are lacking but we note that the auditory sensitivity of the closely related owl monkey, *Aotus trivirgatus*, is similar with that of humans at 500 Hz (*Nummela, 2017*). In general, Neotropical monkeys are most sensitive to sound above 2 kHz (reviewed in *Heffner, 2004*; *Nummela, 2017*), suggesting that roadway traffic may be less disturbing for them. However, traffic noise covers a broad frequency spectrum and sound energy remains substantial up to 10 kHz. Small-headed Neotropical monkeys exhibit their best frequency at ∼8 kHz (*Aotus*, *Callithrix*) and ∼12 kHz (*Saimiri*), which is much higher than the human best frequency (4 kHz) at a similar threshold of loudness (*Heffner, 2004*). For these monkeys, traffic noise in the high frequency range

might represent a source of disturbance. At this point, we can only speculate until studies on the hearing abilities of titi monkeys become available.

## Behavioral changes associated with a gradient of noise levels

Interestingly, the titis' activity budgets seemed to be affected by the anthropogenic noise gradient, as indicated by a negative interaction between the noise levels and adult pair locomotion. The behavioral change occurred within a gradient of minimum sound pressures ranging approximately from 44 dB(A) to 52 dB(A). These noise levels stand above the coterminous USA range for natural sounds devoid of human influence (24–40 dB LAeq; *Shannon et al., 2016*) and approximate those reported at sites close to and far from an open-cast mine in Brazil (*Duarte et al., 2015*). In fact, the mining noise was loud enough to impact the calling patterns of black-fronted titi monkeys (*Duarte et al., 2017*). Correlative changes in songbird vocal activity have also been observed along a noise gradient within an urban park (*Díaz, Parra & Gallardo, 2011*). Whether this noise-related change in the titis' locomotion results from anthropogenic disturbance or is fortuitous (i.e., correlation without causation) remains to be explored. Although little is known about the consequences of sleep deficits in free-ranging nonhuman primates (*Reinhardt et al., 2019*), one possibility is that nocturnal traffic noise might disrupt sleep patterns, more so in adults than their putative offspring. Adult pairs experiencing sleep disruption near the highway would then compensate with longer or more frequent naps during the day. In female frogs, traffic noise significantly increases tonic immobility response with concomitant elevation of plasma corticosterone concentrations (*Tennessen, Parks & Langkilde, 2014*).

The titis' alarm response to a mannequin was weaker in groups most exposed to anthropic disturbance (G1, G2), thus revealing a potential habituation to human presence. However, contrary to our hypothesis, their response did not vary according to the degree of human disturbance in each of the three areas of the park. Of the six study groups, the one that resided in the area with moderate human disturbance (G3 in area 2, where human intrusion was less likely due to dense bamboo vegetation) exhibited the strongest response to the mannequin. This group carried an un-weaned infant, which could explain their strong reaction to the mannequin. In all successful trials, call B was the predominant call type emitted by the titis during the first minute after detecting the mannequin. Previous studies of black-fronted titis' alarm calls (*Cäsar et al., 2012*; *Cäsar et al., 2013*) reported that call B is given in a variety of situations, the context of which is specified by the position of the call in the overall sequence. For instance, the appearance of a terrestrial predator or any kind of disturbance on the ground will trigger call B, which is also emitted while the titis are descending or foraging close to the ground. Of interest, non-habituated groups of black-fronted titis also produced call B in response to human presence (*Cäsar et al., 2012*). Although our subjects might have viewed the mannequin as a human, performing control tests with a scrambled, desarticulated mannequin could have shed light on this issue.

## Relationship between the noise gradient and fecal cortisol

Contrary to our expectation, fecal cortisol levels measured in the six study groups were unrelated to the noise gradient. It is conceivable that the absence of increased fecal cortisol

in groups residing near the highway was due to the rather low statistical power of our data and the variable time in sample collection, thus limiting the scope of our study. Although time of day had no effect on cortisol titers, a more adequate sampling might have shown the presence of a diurnal cycle in cortisol secretion. Such pattern has been reported for the squirrel monkey and common marmoset, revealing a diurnal decrease, both in circulating and salivary cortisol (*Coe & Levine, 1995*; *Cross & Rogers, 2004*). For fecal cortisol, however, higher levels were measured in the afternoon, not in the morning, thus reflecting the time necessary for hormonal excretion (*Sousa & Ziegler, 1998*; but see *Ferreira Raminelli et al., 2001*). In primates, the lag time for the appearance of fecal cortisol is affected by body size, diet and gut transit time, with the response occurring within 4–48 h of a stressful event (*Rangel-Negrín et al., 2009*; *Wark et al., 2016*; *Chen et al., 2017*). Environmental stressors other than noise could also have confounded cortisol titers in our subjects. For instance, we did not take into account group size, inter-group social conflicts, food resources and predation risk, all of which are potential sources of chronic stress that could potentially override subtler variations induced by the noise gradient. Further work involving a larger sample of feces will be needed to clarify these points.

Fecal glucocorticoids are expected to be less concentrated than circulating hormones, but the levels we measured in *P. donacophilus* are ~200 times lower than plasma cortisol reported for *P. cupreus* in captivity (*Hennessy et al., 1995*; *Ragen et al., 2013*; *Fisher-Phelps et al., 2015*; *Hostetler et al., 2016*) and about ~100 times lower than fecal cortisol measured in other Neotropical monkeys (*Albuquerque et al., 2001*; *Ange-van Heugten et al., 2009*; *Wheeler et al., 2013*; *Cantarelli et al., 2017*; *Price et al., 2019*). Our fresh fecal samples were immersed in aqueous alcohol solution and then transferred to a freezer. Storage in alcohol prevents degradation of hormonal metabolites by bacterial metabolism, even at room temperature (*Hodges & Heistermann, 2011*; *Kalbitzer & Heistermann, 2013*). Nevertheless, we cannot exclude the possibility that the dilution might have caused the loss of signal, prior to analysis. Lastly, our choice of using an EIA kit normally used for salivary cortisol might explain its weak reactivity to fecal extracts, thus resulting in low hormone concentrations in all individuals, regardless of age, sex, time-of-day and home range location relative to the highway. However, none of the most relevant problems associated with EIA (*Brown, Walker & Steinman, 2005*) was identified in our assay protocol. In fact, the presence of two higher cortisol values obtained from one adult female shows that the Salimetrics kit was working.

Admittedly, along with our chemical validation, it would have been useful to perform a physiological validation with some type of acute stress or with a pituitary trophic hormone (ACTH) stimulant to induce maximal glucocorticoid release and thus determine the biological range of cortisol concentration in our population (*Brown et al., 2004*; *Wheeler et al., 2013*; *Behringer & Deschner, 2017*). Such a test, however, required the capture and restraint of an animal, which was not possible during the study period. Finally, three groups of titis had weaned infants but it is very unlikely that the females in those groups— including the one that provided the highest cortisol titers—were pregnant at the time of sample collection. In female tamarins, fecal cortisol has been shown to rise in late pregnancy (*Bales et al., 2005*; *Price et al., 2019*).

### Relevance to primate conservation

To our knowledge, we conducted the first road impact study mediated by traffic noise in titi monkeys. Consistent with a previous study (*Dingess, 2013*), the titis' home ranges at Yvaga Guazú overlap substantially and their size is among the smallest reported for the subfamily Callicebinae (*Norconk, 2007*; *Bicca-Marques & Heymann, 2013*; *Huck, Di Fiore & Fernandez-Duque, 2020*). Furthermore, the titis' travel speed (78–95 m/h) in the park was quite low compared with that (300–420 m/h) reported for the masked titi monkey, *Callicebus nigrifrons*, in a much larger patch of Atlantic forest (*Nagy-Reis & Seitz, 2017*). This is not surprising given the heavy deforestation around the park with little opportunity for family group dispersion. The titis, however, spent most of their time resting, which accords with activity budgets of other callicebine monkeys (*Caselli & Setz, 2011*; *Bicca-Marques & Heymann, 2013*; *Kulp & Heymann, 2015*; *Van der Speld, Bello & Hebard, 2017*; *Dolotovskaya & Heymann, 2020*). These cryptic and shy animals have been reported to thrive in disturbed and/or fragmented forests (*Bicca-Marques & Heymann, 2013*), despite the negative effects of road networks on animal and plant communities (*Gill, Sutherland & Watkinson, 1996*; *Laurance, Goosem & Laurance, 2009*; *Hernani-Lineros, Garcia & Pacheco, 2020*). Evaluating further their tolerance threshold to anthropogenic noise will assist in the conservation and management of their populations (*Rumiz, 2013*; *Shanee et al., 2013*; *Wallace et al., 2013*). Finally, three of our study groups had un-weaned infants suggesting that population growth at Yvaga Guazú was not constrained by the level of human disturbance, an indication that this peri-urban population of titis is able to cope with human disturbance.

## CONCLUSION

Emerging from this study are changes in the titis' behavior in association with increased human disturbance. Concomitantly, our physiological data using fecal cortisol as a proxy for chronic stress are suggestive of the titis' resilience to anthropic perturbation, but do not allow us to draw firm conclusions owing to sampling issues. Though this topic merits further investigation, a major emphasis of this study has been to combine a suite of acoustical, behavioral and hormonal methodologies, supplemented by field experiments. Such conceptual framework is worth pursuing if we are to disentangle the complex interactions between human disturbance, behavior and stress physiology in naturalistic settings. In addition, laboratory studies are needed to elucidate the auditory sensitivity of titi monkeys to anthropic noise and to assess hormonal responses under more controlled conditions, via playback experiments. We hope that the present study will stimulate further research in those directions.

## ACKNOWLEDGEMENTS

Part of the work reported here was executed by LHL to fulfill a Master's degree at the Universidad Mayor Sans Andrés, La Paz (Bolivia). The Museo de Historia Natural Noel Kempff Mercado provided institutional support through Kathia Rivero, Head of Vertebrate Zoology, to obtain research approvals and export permits from the National

Biodiversity Authority. Engineer César Perez Alcover developed the timer for passive acoustic monitoring. For the processing of biological samples, we thank the Plateau de Paléogénomique et Génétique Moléculaire (P2GM), Museum National d'Histoire Naturelle, Paris, France. We are most indebted to Emeritus Professor Michel Kreutzer for insightful comments on an early draft of the manuscript. Special thanks are due to Rebeca Rozenman Attie and Francisco Hübsch, landowners of the Ecological Park of Yvaga Guazú. We also thank Rosely Ligeron Arteaga and Franco Oscar Echenique Robles for field assistance at an early stage of the study.

### Funding

This work was supported by the Academia Nacional de Ciencias de Bolivia/UPSA, Santa Cruz [grant number UPSA-ANCB-SC-01-2017] and the Foundation IDEA WILD awarded the cortisol kit. The funders had no role in study design, data collection and analysis, decision to publish, or preparation of the manuscript.

### Grant Disclosures

The following grant information was disclosed by the authors:
Academia Nacional de Ciencias de Bolivia/UPSA, Santa Cruz: UPSA-ANCB-SC-01-2017.
Foundation IDEA WILD.

### Competing Interests

The authors declare there are no competing interests.

### Author Contributions

- Lucero M. Hernani Lineros and Patrice Adret conceived and designed the experiments, performed the experiments, analyzed the data, prepared figures and/or tables, authored or reviewed drafts of the paper, and approved the final draft.
- Amélie Chimènes performed the experiments, analyzed the data, authored or reviewed drafts of the paper, and approved the final draft.
- Audrey Maille and Kimberly Dingess analyzed the data, authored or reviewed drafts of the paper, and approved the final draft.
- Damián I. Rumiz conceived and designed the experiments, authored or reviewed drafts of the paper, and approved the final draft.

### Animal Ethics

The following information was supplied relating to ethical approvals (i.e., approving body and any reference numbers):

Field experiments were approved by Ministerio de Medio Ambiente y Agua, Dirección General Biodiversidad y Areas Protegidas (DGBAP) MMAyA-VMABCCGDF-DGBAP, No 059/2014

## Field Study Permissions

The following information was supplied relating to field study approvals (i.e., approving body and any reference numbers):

The landowners Rebeca Rozenman Attie and Francisco Hübsch, provided authorization to access the Ecological Park of Yvaga Guazú.

## Data Availability

The raw data is available in the Supplemental File.

## Supplemental Information

Supplemental information for this article can be found online at http://dx.doi.org/10.7717/peerj.10417#supplemental-information.

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

## FURTHER READING

**Estrada A, Garber PA, Rylands AB, Roos C, Fernandez-Duque E, Di Fiore A, Nekaris KA, Nijman V, Heymann EW, Lambert JE, Rovero F, Barelli C, Setchell JM, Gillespie TR, Mittermeier RA, Arregoitia LV, De Guinea M, Gouveia S, Dobrovolski R, Shanee S, Shanee N, Boyle SA, Fuentes A, MacKinnon KC, Amato KR, Meyer ALS, Wich S, Sussman RW, Pan R, Kone I, Li B. 2017.** Impending extinction crisis of the world's primates: why primates matter? *Science Advances* **3**:e1600946 DOI 10.1126/sciadv.1600946.