# Peer review of "Response of Bolivian gray titi monkeys (Plecturocebus donacophilus) to an anthropogenic noise gradient: behavioral and hormonal correlates"

_PeerJ, doi:10.7717/peerj.10417_

## Round 0.1 · original submission · Minor Revisions

I agree with reviewer 3 that you have done a good job with the suggested corrections. I hope that you will make a last effort and add the suggestions that have been included directly in the document. However, some important questions remain to be clarified, such as the variation in cortisol according to day time, raised by reviewer 1. Please clarify this in your methods, and explain how it could influence your results

Reviewer 1 ·

Basic reporting

The authors have done an excellent job with the writing, literature review, and presentation of data.

Experimental design

The researchers have provided a very strong descriptive basis for their study (including identification of individual animals; mapping of home ranges; and measurements of noise pollution), but the experimental (mannequin) and physiological (cortisol) outcome measures are hindered by small samples and too many confounding issues.

Validity of the findings

My main concerns are with the cortisol data. There are a number of factors which may have contributed to the lack of findings here.

1) Time of day. Cortisol varies systematically with time of day. I see no mention of what time of day the samples were collected or any examination of its effects. The possible circadian effects are mentioned in the discussion, why aren’t we given more information on the actual samples? A previous reviewer also asked about time since exposure to the mannequin – this information is also possibly pertinent, and missing from the manuscript.
2) Potential pregnancy. Cortisol is higher in pregnant females. Since three groups had weaned infants, what was the possibility that the female in each of these groups was pregnant? If the time of year makes this unlikely, fine. But if the assay was not detecting pregnant females, then it adds further doubt to how well it was working.
3) Low levels. It is very difficult to evaluate why the measured cortisol levels are so low here, but it is worrying and the authors don’t really have an explanation.
4) The validation is also puzzling. Essentially, they did not validate for parallelism, and the 4 samples that they were using for the validation had “identical” values. This seems odd and unlikely.
5) The authors themselves point out the numerous confounds that may have existed and that were not incorporated into their analyses.

Altogether, I expect that the cortisol measures are just too messy to be reliable, or that something happened to affect the samples.

I also wondered to what extent the authors are trying to assert that the titis viewed the mannequin as a human – which seems to be what they’re suggesting in lines 117-119. How do you know that they weren’t just reacting to a giant novel object? How do these reactions compare to the reactions to an actual human?

Additional comments

Line 401 and Table 3: Should be “Tail-twining” not “tail-twinning”

Reviewer 2 ·

Basic reporting

L36: Where are the predictions for the effect of noise on group movement and behaviour?
L53: This sentence is too long, try to break it up a little so its easier to follow.
L58: I’m not sure that light and noise pollution comes under the bracket of human-wildlife conflict. A human-wildlife conflict normally occurs when an animal poses a threat to humans leading to persecution of the animal and a conflict about the best way to resolve the issue. I would remove this sentence to think of an alternative framework that this paper fits within.
L66: “which is considered as one of the most pervasive acoustic perturbations on Earth” - what specifically is considered this? The layout of this sentence is slightly confusing.
L79: I don’t think that increased vigilance is a way to stay away from the source of stress - maybe rephrase that
L89: “field studies of Neotropical primates are only beginning to emerge in few taxa” - is this in general or in terms of noise pollution? You actually cite a lot of papers on this topic so can you argue that this is an emerging research topic?
L101: I’m not sure this sentence on edge effects is really relevant here.
L117: “In addition, to assess whether the monkeys” - should have an apostrophe after monkeys.
L140: Maybe provide more description of how you subjectively subdivided the study area.
L190: You could specify that you have three recorders at the highway to clarify how you have 17 recording sites.
L206: The RMS index is very well explained and would be clear to non-experts what this index represents and how you would interpret that data. I think it would be good to give a similar explanation of the M index.
Line 229: Not sure about the use of the word “his”.
L542: Sentence beginning “Our findings concur with previous studies of black-fronted titis” - this sentence doesn’t really add anything or explain what finding your study supports. You state that call B is produced in a variety of situations but fail to give any examples.
L565: Should say “We think it is unlikely”.


Figure 1A: I find it difficult to see where the park is located on the map. Maybe zoom out slightly and include an outline of the park.
Figure 2A, B and C: Why are these graphs reported for the RMS index and not the M index?
L320: What do you mean by extreme values? Maybe explain a little more why you are including certain things in your model.

Experimental design

L171: Why did you orient the sound level meter towards the highway and towards Campo Verde?
L172: How did you sample all of the stations on a grid 5 times a day? Did you have 9 people?
L257: How did you decide on the 1m and 1.5m to 7m cut-offs for loose and close associations?
For the statistical analysis, I would recommend mixed models with individual values instead of averaging across all of your variables e.g. averaging across cortisol values and noise values.

You should describe the different call types in the methods, how you distinguish them and what they represent.

Validity of the findings

Could there be an effect of habituation by the observer in the more remote groups?

Could the structure of the forest or physical disturbance explain these patterns instead of noise pollution?

L576: I would round this paragraph off by referring back to your study. What does your research have to contribute to this conversation?

Additional comments

No comment

Reviewer 3 ·

Basic reporting

The authors did a good job improving the paper but it still needs some adjustments. I made some suggestions on how to improve the abstract and the discussion directly in the word file.

Experimental design

No problems regarding the experimental design.

Validity of the findings

Conclusions are well stated, linked to original research question.

Additional comments

The authors did a good job improving the paper but it still needs some adjustments. I have some suggestions on how to improve the abstract and the discussion and you can find them directly in the word file.

Annotated reviews are not available for download in order to protect the identity of reviewers who chose to remain anonymous.

---

## Round 0.2 · accepted · Accept

Dear authors, I think you have finally done a good review including all suggestions and explaining all reviewers' concerns! congratulations!

Reviewer 1 ·

Basic reporting

All good here

Experimental design

Experimental design is solid

Validity of the findings

I still have some concerns over the fecal cortisol analysis; however, the authors do a good job in the discussion of talking about the limitations.